A preliminary assessment of population genetic structure of the common vampire bat (Desmodus rotundus) in Colombia

Van de Vuurst Paige 1 2 3
Cifuentes-Rincon Analorena 1
Bertke Andrea S. 3 4
Soler-Tovar Diego 1 5
Reyes-Amaya Nicolás 6
Rodriguez Arévalo Fabiola 7
Cárdenas Hincapié Julieth Stella 8
Rivera-Monroy Jhon 9
Escobar Luis E. escobar1@vt.edu 1 3 10 11
Hallerman Eric 1
1 Department of Fish and Wildlife Conservation, Virginia Polytechnic Institute and State University (Virginia Tech) , Blacksburg , VA , United States of America
2 Translational Biology, Medicine, and Health Program, Virginia Polytechnic Institute and State University (Virginia Tech) , Blacksburg , VA , United States of America
3 Center for Emerging Zoonotic and Arthropod-borne Pathogens, Virginia Polytechnic Institute and State University (Virginia Tech) , Blacksburg , VA , United States of America
4 Population Health Sciences, Virginia Maryland College of Veterinary Medicine, Virginia Polytechnic Institute and State University (Virginia Tech) , Blacksburg , VA , United States of America
5 Epidemiology and Public Health Research Group, Faculty of Agricultural Sciences, Universidad de La Salle - Bogotá , Colombia
6 Colección de Mamíferos, Instituto de Investigación de Recursos Biológicos Alexander von Humboldt , Bogota , Colombia
7 Laboratorio Nacional de Diagnóstico Veterinario (LNDV), Instituto Colombiano Agropecuario (ICA) , Bogota , Colombia
8 Museo de La Salle (MLS), Universidad de La Salle - Santafé de Bogotá , Bogota , Colombia
9 Laboratorio Instrumental de Alta Complejidad (LIAC), Universidad de La Salle - Santafé de Bogotá , Bogota , Colombia
10 Kellogg Center for Philosophy, Politics, and Economics, Virginia Polytechnic Institute and State University (Virginia Tech) , Blacksburg , VA , United States of America
11 Global Change Center, Virginia Polytechnic Institute and State University (Virginia Tech) , Blacksburg , VA , United States of America
Nelson David
Electronic publication date: 2025 Nov 10
Publication date: 2025
Volume: 13
Electronic Location ID: e20306
Received 2025 May 23; Accepted 2025 Oct 7
Copyright: ©2025 Van de Vuurst et al.
Copyright year: 2025
Copyright holder: Van de Vuurst et al.
License: This is an open access article distributed under the terms of the Creative Commons Attribution License, which permits unrestricted use, distribution, reproduction and adaptation in any medium and for any purpose provided that it is properly attributed. For attribution, the original author(s), title, publication source (PeerJ) and either DOI or URL of the article must be cited.
License URL: https://creativecommons.org/licenses/by/4.0/

Keywords: Desmodus rotundus, Genetic variation, Populations dynamics, Elevation, Phylogeography

Funding: National Science Foundation under CAREER (2235295) and HEGS (2116748) awards National Institute of Allergy and Infectious Diseases of the National Institutes of Health K01AI168452 Virginia Tech’s Institute for Critical Technology and Applied Science, Pandemic Prediction and Prevention Destination Area, and the Center for Emerging, Zoonotic, and Arthropod-borne Pathogens Chinese Academy of Sciences PIFI project 2024PVC0085 Virginia Agricultural Experiment Station with support from the U.S. Department of Agriculture National Institute for Food and Agriculture This study was supported by the National Science Foundation under CAREER (2235295) and HEGS (2116748) awards, by the National Institute of Allergy and Infectious Diseases of the National Institutes of Health under award number K01AI168452, and by seed grants from Virginia Tech’s Institute for Critical Technology and Applied Science, Pandemic Prediction and Prevention Destination Area, and the Center for Emerging, Zoonotic, and Arthropod-borne Pathogens. Luis Escobar was supported by the Chinese Academy of Sciences PIFI project 2024PVC0085. The participation of Eric Hallerman was supported through the Virginia Agricultural Experiment Station with support from the U.S. Department of Agriculture National Institute for Food and Agriculture. There was no additional external funding received for this study. The funders had no role in study design, data collection and analysis, decision to publish, or preparation of the manuscript.

==============================
Rabies virus (RABV) is a neglected tropical pathogen in Latin America predominantly transmitted to mammals by the common vampire bat (Desmodus rotundus). Transmission of RABV among D. rotundus individuals and colonies is a function of individual dispersal between colonies, patterns of which can be inferred from population genetic structure. Nevertheless, a baseline assessment of population genetic structure among D. rotundus individuals has been lacking for some areas of South America, including Colombia, where RABV has impacted some areas more heavily than others. To assess individual dispersal and hence population structure of D. rotundus across heterogenous landscapes in Colombia, we conducted a cross-elevational assessment of population genetic variation using nuclear microsatellite DNA markers. We quantified genetic variance and geographic distribution of genetically clustered D. rotundus individuals across the landscape of Colombia with reference to a comparator group of individuals from Mexico. We found population-level differentiation and genetic structure within our collection of samples, and we inferred patterns of dispersal and genetically effective migration between D. rotundus populations. Analysis of molecular variance (AMOVA) showed considerable variation among inferred populations in Colombia (14.9% of genetic covariance, df = 2, Sum of Squares = 164.9, Sigma = 1.28, ϕ = 0.15, p = 0.01), with an associated G′ST of 0.34. Direct migrant identification suggested 15 likely first-generation migrants among sites. We found that there were no statistically significant differences between the landscapes occupied by the inferred populations, though our limited sampling size suggests a trend toward differences in relation to elevation (t = 1.91, df = 71.72, p = 0.06). These results indicate that D. rotundus is mobile within the region, potentially contributing to RABV transmission among colonies. Our results support previous hypotheses ecological resistance-mediated patterns of dispersal for D. rotundus, and inform future research on the role of genetic connectivity in RABV transmission among bat colonies.

Introduction

Rabies virus (RABV) is a prevalent and highly impactful pathogen (Anderson et al., 2014; Velasco-Villa et al., 2017) predominantly spread from bats to other mammalian species in Latin America by the common vampire bat (Desmodus rotundus) (Barquez et al., 2015). RABV outbreaks in humans and livestock regularly occur in Latin America (Stoner-Duncan, Streicker & Tedeschi, 2014; Soler-Tovar & Escobar, 2025), where human infections regularly follow outbreaks in livestock (Meske et al., 2021). Spread of RABV among D. rotundus colonies is thought to be a function of individual dispersal and migration patterns between populations (Blackwood et al., 2013; Benavides, Valderrama & Streicker, 2016; Megid et al., 2021) patterns of which can be inferred from population genetic structure. An understanding of basic population genetics and geographic structure of D. rotundus is beneficial for local stakeholders to mitigate RABV’s impact on humans and animals. Assessments of populations structure along landscapes is further justified in areas with pronounced landscape heterogeneity, such as that found in Latin America. Landscape heterogeneity impacts the spatial distribution and overall health of wildlife species and infectious disease dynamics (Biek & Real, 2010; García-Morales, Badano & Moreno, 2013; Becker et al., 2019). Furthermore, assessing population genetic structure can bypass the need for lengthy telemetry, mark-recapture, or longitudinal serology-based assessments of bat movement that would otherwise be needed to elucidate possible transmission routs between populations (Blackwood et al., 2013; Mara, Wikelski & Dechmann, 2014; Schorr et al., 2014; Streicker & Allgeier, 2016; Megid et al., 2021). To assess how landscape heterogeneity may influence population structure of D. rotundus, we conducted a cross-elevational assessment of population genetic structure for this species in Colombia using nuclear microsatellite DNA markers.

Colombia is a country with large environmental gradients within a relatively small area, and has endemic circulation of RABV (Fernando et al., 2017; Cediel, 2020; Rojas-Sereno et al., 2022), which is regularly monitored by the Epidemiological Information and Surveillance System of the Instituto Colombiano Agropecuario (ICA) (Instituto Colombiano Agropecuario (ICA), 2019; Instituto Colombiano Agropecuario (ICA), 2020; Bonilla-Aldana et al., 2022). Bovine rabies in Colombia shows marked regional variation and evolving outbreak dynamics. Bonilla-Aldana et al. (2022) reported 4,888 confirmed cases between 2005 and 2019 in Colombia, with a significant decline after 2014 and a notable concentration of cases in the Department of Sucre (Caribbean Region, North Colombia). Other research has citied low vaccination coverage and implementation difficulties as potential drivers for RABV outbreaks (Álvarez, Buitrago & Sáenz, 2014). Across the last five decades, 2,858 outbreaks of RABV have been reported in Colombia in cattle alone (Soler-Tovar & Escobar, 2025). Furthermore, Colombia has experienced an expansion of D. rotundus-associated RABV into novel areas, and an increase in landscape change since the cessation of armed conflict in 2016 (Clerici et al., 2020; Rojas-Sereno et al., 2022; Bonilla-Aldana et al., 2022). As a result, there is a need for a population genetics assessment for vampire bats in this region, especially within the context of landscape heterogeneity. Elevation can be used as a valuable proxy for environmental variation, since changes in altitude often correlate with changes in numerous environmental factors, such as temperature, vegetation composition, and humidity, making elevation a useful metric for studying landscape heterogeneity impacts (González, Willig & Waide, 2013). We therefore evaluated genetic diversity, population structure and its spatial distribution among sampling locations across an elevational gradient in the Colombian Andes. Our objective was to assess the extent to which population genetic diversity and structure of D. rotundus populations varied across space and were associated with environmental factors, which could be used to infer RABV transmission dynamics. This information could be useful to target RABV surveillance and pre-exposure human and livestock vaccination efforts.

Materials and Methods

Experimental design

To assess spatial patterns of genetic diversity and structure in D. rotundus populations, and to test their association with environmental factors relevant to rabies virus (RABV) transmission, we implemented a multi-step population genetic approach. This included: (1) broad-scale field sampling across an elevational gradient in Colombia, (2) genotyping at highly polymorphic microsatellite loci, (3) quantifying within- and among population genetic diversity and structure, and (4) testing the relationships between genetic metrics and environmental variables including geography, elevation, climate, and landscape features at the locations of capture.

We utilized Colombia’s dynamic topography and ecological diversity to select sampling locations across an elevational gradient in the Andes Mountains, under the assumption that elevation can be used as a reliable proxy for a variety of environmental factors (González, Willig & Waide, 2013; Lyons & Kozak, 2019). Thus, sample sites were selected to encompass elevations from low (<500 m), to moderate (500–1,000 m), to high (>1,000 m) to ensure that D. rotundus individuals were captured from a diversity of environments. Sites were also selected to minimize potential spatial autocorrelation and to reflect home-range distances of D. rotundus (McNab, 1973; Trajano, 1996), and hence were spaced at least 20 km apart. Elevational categories (i.e., low, moderate, and high) were utilized during site selection, but not in subsequent analyses. That is, all assessments were made independent of elevational category so as to not skew any potential relationships identified in subsequent analyses.

We obtained tissue samples from 81 D. rotundus individuals from Colombia in RABV-endemic areas during June and July of 2022 and 2023 (Table S1). Complementarily, five samples from a previous study collected between 2019 and 2022 were provided by the Instituto de Investigación de Recursos Biológicos Alexander von Humboldt (Bogota, Colombia) for the purposes of this study (Table S1), totaling 86 D. rotundus individuals from Colombia. As a genetic comparator, we included six D. rotundus individuals from western Mexico (Piaggio, Johnson & Perkins, 2008) (Table S1, Fig. 1) to contextualize population divergence. Environmental variables used for downstream associations with genetic structure included elevation (m), isothermality (average diurnal temperature range in ∘C divided by annual temperature range in ∘C), minimum temperature of the coldest month (∘C), annual average precipitation (mm), precipitation seasonality (coefficient of variance) and landscape features such as associated land use and vegetation cover (i.e., cattle density or vegetation indices present at the location of capture) for each sample site. Environmental variables were derived from the NASA Socioeconomic Data and Applications Center (SEDAC) (NASA Earth Data, 2022), Gridded Livestock of the World Database (Food and Agriculture Organization of the United Nations, 2022), and the WorldGrids Archived database (Hengl, 2018) at 1-km spatial resolution. Variables were selected based on their ecological importance to D. rotundus and RABV spillover risk as established in previous research (Lyman & Wimsatt, 1966; Zarza et al., 2017; Mantovan et al., 2022; Magalhães et al., 2023; Van de Vuurst et al., 2023; Jones et al., 2024). Variable temporal range of the climate and landscape data were from 2016–2020.

Figure 1 Sample sites for tissue collection.

(A) Locations of sample sites for D.rotundus individuals in Colombia. All sampling sites were at least 20 km from one another. Greatest distance between sampling sites was 669 km, originating from the sampling location of El Porvenir, Córdoba (far north Colombia), which was provided by the Humbolt Institute of Colombia. (B) Sites of collection for D. rotundus samples from Mexico provided by A. Piaggio. (C) Colombia sample sites overlayed over a map of Isothermality (average diurnal temperature range divided by Annual temperature range) to denote bioclimatic differences in sampling sites. Geographic coordinates for sampling sites are provided in Table S1. Maps created using ArcGIS Pro software version 2.5 with shape files from DIVA-GIS and topographic maps from USGS (Hijmans, 2017; Esri, Redlands, CA, USA, 2019). Isothermality data were sourced from the WorldGrids Archived database (Hengl, 2018) .

Bat capture and tissue collection

Desmodus rotundus individuals were collected using mist nets (12 × 2.6 m, mesh size 36 mm) for at least five consecutive hours after sunset targeting periods of peak activity during the waning moon phase (Zeppelini, Azeredo & Lopez, 2019). Animal handling followed protocols adhering to recognized bat handling guidelines of the American Society of Mammalogists (Sikes, 2016), as approved by the Institutional Animal Care and Use Committee at Virginia Tech (IACUC approval #21-138). Approval was also obtained from the Institutional Care and Use Committee of the Universidad de La Salle (IACUC approval #087). Our sampling protocol abided by legal regulations of Colombia, including the National Statute for the Protection of Animals, Resolution 8430 of the Ministry of Health: scientific, technical, and administrative standards for health research, the Code of Ethics for the professional practice of veterinary medicine, and Law 1774, amending the Civil Code, Law 84 of 1989, the Penal Code, and the Code of Criminal Procedure. Capture of bats and data collection in Colombia were carried out under environmental permit Resolución 1473 de 2014 of the Autoridad Nacional de Licencias Ambientales (ANLA) (i.e., UniSalle Wild Species Collection Framework Permit). Both capture-and-release sampling and lethal sampling were conducted to collect the tissues analyzed in this study. Tissue samples were collected via pectoralis muscle dissection from individuals that were lethally sampled as part of an independent bat virome study aimed at reconstructing the virosphere of bats in the tropics (IACUC approval #21-138). For released individuals, tissue samples were collected via a 1-mm wing punch. All individuals that were captured during netting were sampled without discrimination based on age class, sex, or location.

To minimize stress and potential harm for captured individuals, bats were removed from nets and handled gently using clean cloth bags and restrained carefully by trained personnel. For euthanasia, bats were carefully placed inside a sealed Ziploc-type plastic bag containing a cotton pad soaked with two mL of USP-grade isoflurane, positioned to prevent direct contact with the animal. Bats were continuously observed for signs of anesthesia (i.e., decreased responsiveness to stimuli, loss of righting reflex, and slowed respiration). Once deep anesthesia was confirmed via absence of reflexes, cardiac puncture was conducted to induce exsanguination, followed by cervical dislocation to ensure death (Sikes, 2016). Death was confirmed by the absence of heartbeat, respiration, and the presence of fixed, dilated pupils. All samples were preserved at −20 °C and stored long-term at −80 °C at the Universidad de La Salle in Bogotá, Colombia.

DNA extraction and microsatellite genotyping

DNA was extracted using the DNeasy Blood & Tissue Kit (Qiagen) and quantified with a NanoDrop 2000 spectrophotometer (Thermo Fisher Scientific) (Qiagen, 2023) at the Universidad de La Salle in Bogota, Colombia. All extractions had a yield of 50 ng/µL or greater. For each individual, we then amplified 12 nuclear microsatellite loci previously developed by Piaggio, Johnson & Perkins (2008) using Promega GoTaq Gold PCR protocols (Promega, 2019). Each 25-µl PCR reaction consisted of 1.0 µl 25 mM MgCl2, 5.0 µl of 5X colorless GoTaq Flexi buffer (Promega, Madison, WI, USA), 0.5 µl of 10 mM PCR nucleotide mix, 1.0 µl of 10 µM forward and reverse primers, 0.125 µl of GoTaq DNA polymerase, 2.0 µl of template DNA (<0.25 µg/25 µl), and 14.375 µl of nuclease-free water. PCR amplification was conducted using a T-100 thermocycler (BioRad, Hercules, CA, USA) with a profile consisting of initial denaturation at 95 °C for 10 min; followed by 35 cycles of 94 °C for 30s, 52 °C for 45s, and 72 °C for 45s; and a final extension at 72 °C for 10 min. Samples then were held at 12 °C for at least 15 min prior to being placed in storage at 2 °C (Romero-Nava, León-Paniagua & Ortega, 2014). PCR products were sent to the Cornell University Institute of Biotechnology Resource Center (Ithaca, NY) for fragment analysis. We used the Fragman package (version 1.0.9) (Covarrubias-Pazaran et al., 2016) in R statistical software version 4.1.0 (R Core Team, 2021) to automate peak calls for determining amplification fragment sizes for alleles at each of the 12 loci for each D. rotundus individual. Allele lengths were confirmed via a visual inspection of peaks for each locus in GeneMarker version 2.6.3 software (SoftGenetics, 2018). We used Brookfield (Brookfield, 1996; Campagne et al., 2012) and Van Oosterhout methods (Van Oosterhout et al., 2004; Van Oosterhout, Weetman & Hutchinson, 2006; The Earth and Life Systems Alliance, 2023) through the PopGenReport package version 3.1 (Adamack & Gruber, 2014) in R to identify and exclude loci with high null allele frequencies (>0.5), ensuring data reliability.

Population genetics analysis

To assess how genetic variation was structured across space and potentially linked to environmental factors, we conducted the following sequential analyses:

Population structure and clustering.

To infer genetic clusters without prior assumption, we applied two complementary approaches: Discriminant Analysis of Principal Components (DAPC) (Jombart, Devillard & Balloux, 2010) using sequential K-means of the number of populations or conformance of genotype frequencies to Hardy-Weinberg equilibrium (HWE) expectations (Jombart, 2015; Desvars-Larrive et al., 2019), optimized using Bayesian information criterion (BIC) for each increasing number of K. We also applied Bayesian Clustering via STRUCTURE software (Pritchard, Stephens & Donnelly, 2000) applying the log-likelihood approach of Pritchard, Stephens & Donnelly (2000) and the delta K (δK) method of Evanno, Regnaut & Goudet (2005) to identify the best-supported value of K. We applied the poppr package version 2.9.6 (Kamvar, Tabima & Grunwald, 2014) to conduct analysis of molecular variance (AMOVA) within individuals, among individuals within clusters, and between clusters. These methods allowed us to identify putative populations and assess admixture, forming the basis for all population-specific analyses.

Genetic diversity and differentiation.

For each cluster identified by DAPC, we used hierfstat version 0.5-11 (Jerome et al., 2022) and dartRverse version 2.0 (Mijangos et al., 2022) to calculate:

1. Allelic richness, observed and expected heterozygosity, and inbreeding coefficient (FIS, (Weir & Cockerham, 1984).

2. Pairwise FST and G′ST (Yang, 1998; Hedrick, 2005) as measures of genetic differentiation using dartRverse. Statistical significance was evaluated using 1,000 permutations, with p-values < 0.05 indicating significant genetic differentiation.

3. Deviations from Hardy-Weinberg Equilibrium (HWE) using the poppr package (Kamvar, Tabima & Grunwald, 2014)

Effective population size (Ne) was estimated using NeEstimator (Do et al., 2014) with the Waples correction (Waples & Do, 2010; Waples, Larson & Waples, 2016) to infer local demographic stability.

Isolation-by-distance and sex-based dispersal.

To evaluate whether genetic differentiation was driven by distance or environmental factors, we conducted a Mantel (1967) test using Euclidean distance, FST, and G′ST in the PopGenReport and dartRverse packages (Adamack & Gruber, 2014; Mijangos et al., 2022). A Mantel test also was used to assess isolation-by-distance among our Colombia samples, and the relationship between the elevation of sample sites and genetic distance. Statistical significance was assessed using 10,000 permutations, with significance thresholds set at p < 0.05.

Additionally, sex-specific spatial autocorrelation analyses was conducted to test for fine-scale structure and sex-biased dispersal using FST of females vs males, and by comparing genetic distance and geographic distance of each sex (Banks & Peakall, 2012). Autocorrelation values were calculated using Euclidean distance matrices of genetic and geographic distances and visualized using the dartR package (Mijangos et al., 2022). We then estimated 95% bootstrap confidence intervals (100 bootstrap replications) around the r values for each sex at different distance-classes based on our sample sites. Assessment of dispersal by sex for each independent population was not possible due to limited power for Population Three, where only five females were captured. As a result, sex-biased dispersal was conducted using all pooled samples from Colombia.

Inference of migration.

To distinguish between historical migration and contemporary movement, we estimated equilibrium migration rates (m) using the FST formula (Wright, 1951; Zhivotovsky, 2015). Values were interpreted based on conventional thresholds (e.g., Nm >1 indicates historic migration between populations). Also, we identified putative recent first-generation migrants using the Bayesian criterion in GeneClass2.0 (Piry et al., 2004) with 10,000 simulated individuals and a significance threshold of p < 0.01. identifying individuals likely to have moved among populations (Piry et al., 2004). This dual approach allowed us to infer both long-term and recent connectivity among D. rotundus populations in Colombia.

Linking genetics and environment.

Finally, we tested for associations between genetic structure and environmental characteristics, both independently and simultaneously, using multiple t-tests and a multivariate analysis of variance (MANOVA) test. For the purpose of this comparison, the comparator group from Mexico was not considered. This allowed us to assess whether environmental variables differed between populations within Colombia without confounding the comparison with data from Mexico. Populations were compared for differences in elevation (m), isothermality (average diurnal temperature range in ∘C divided by annual temperature range in ∘C), minimum temperature of the coldest month (∘C), annual average precipitation (mm), precipitation seasonality (coefficient of variance), cattle density (individuals per km2), human population density, and annual average enhanced vegetation index (EVI) (factors with potential ecological relevance to RABV exposure and transmission risk). We confirmed that the selected environmental characteristics varied across the elevational gradient used for sample site selection using a multiple linear regression model. A p-value < 0.05 was considered statistically significant. We utilized the rstatix package (version 0.7.2) and base R to complete these analyses in R statistical software version 4.1.1 (R Core Team, 2021; Kassambra, 2023).

Results

Sample demographics and genetic characteristics

Genetic variation within a total of 92 D. rotundus individual samples was assessed, 86 from Colombia and six from Mexico. Of the samples collected in Colombia, 79 had known sex, comprising 22 females and 57 males. Forty-two of the sampled individuals were adults, two were sub-adults, six were juveniles, and the remaining 11 were of undetermined age. No individuals sampled were visibly diseased, but pathogen testing on these individuals has yet to be completed. All microsatellite loci proved polymorphic, with numbers of alleles per locus ranging from three to 16. Private alleles, (i.e., alleles found only in specific populations or specific sampling sites) (Petit, Mousadik & Pons, 1998; Szpiech & Rosenberg, 2011), were observed in all populations (Table 1). Evidence of divergence of genotype frequencies from HWE expectations was present for sampling sites Agua de Dios (Cundinamarca), Puente Quetame (Cundinamarca), and Pipiral (Meta) (Fig. 1, Table 1, Table S1) (p = 0.001, 0.03, and 0.01 respectively). All microsatellite loci had null allele frequencies below 0.35, except for locus Dero_H02F_C03R (Fig. S1), data for which was excluded from subsequent analysis. While usual criterion for null allele exclusion is 0.2, this stringent of a threshold would have eliminated over half the loci included in this analysis (Fig. S1). We chose to utilize 0.35 as our threshold in order to retain an optimum number of loci, and based on the understanding that microsatellite loci affected by null alleles are not thought to affect the overall outcome of assignment testing (Carlsson, 2008).

Table 1 Descriptive metrics of populations and sampling sites.

Populations (Pops) represent clusters identified by DAPC analysis. Sampling sites represent locations of collection. N indicates the number of individuals collected at each site. We utilized multiple R packages to quantify the mean number of alleles per locus, private alleles per locus (alleles only found in specific populations or subpopulations), allelic richness, observed (Ho) and expected (He) heterozygosity, significance of divergence from Hardy-Weinberg equilibrium (HWE), and effective population size (Ne) of each cluster identified by DAPC (right column under each metric) and for each sampling site (left column under each metric). Ne was estimated using the Waples correction (Waples & Do, 2010; Waples, Larson & Waples, 2016) to mitigate downward bias. Inf indicates that Ne was estimated as infinite, likely owing to small sample size and related lack of heterozygotes. There was evidence of divergence from HWE for sampling sites Agua de Dios, Pipiral, and Puente Quetame.

Pops	Sampling sites (Department)	N	Elevation (m)	Mean alleles per locus	Private alleles per locus	Allele richness	H o	H e	Departure from HWE	N e	
1	Tamaulipas	4	623.0	2.91	3.90	10	24	1.56	4.18	0.55	0.52	0.56	0.66	0.55	0.53	0.3	Inf	
Nuevo Leon	2	1200.0	2.36	7	1.68	0.55	0.81	0.36	Inf	
2	Agua de Dios (Cundinamarca)	18	407.8	6.45	7.36	4	11	1.65	4.91	0.40	0.37	0.66	0.67	0.001	0.25	20.8	16.9	
Los Araguatos (Arauca)	1	127.0	1.09	1	NA	0.20	NA	0.86	Inf	
Yopal (Casanare)	1	151.0	1.27	0	1.27	0.27	NA	0.81	In.	
Chaparral Sites 1 & 2 (Tolima)	9	615.0 & 654.0	4.45	1	1.64	0.47	0.65	0.24	Inf	
Coello (Tolima)	7	439.2	3.09	0	1.67	0.48	0.72	0.49	Inf	
Medina (Cundinamarca)	1	760.0	1.36	0	1.36	0.36	NA	0.75	Inf	
Pipiral (Meta)	5	965.3	4.18	5	1.73	0.35	0.79	0.01	Inf	
3	Agua de Dios (Cundinamarca)	22	407.8	6.50	7.09	4	11	1.65	4.49	0.40	0.4	0.66	0.64	0.01	0.23	20.8	41.6	
El Porvenir (Córdoba)	1	36.2	1.36	0	1.36	0.36	NA	0.75	Inf	
Ibagué (Tolima)	5	2198.9	3.55	0	1.64	0.35	0.68	0.36	Inf	
Piedras (Tolima)	6	619.1	3.45	1	1.55	0.33	0.58	0.29	0.8	
Puente Quetame (Cundinamarca)	7	1890.0	3.82	2	1.63	0.39	0.65	0.03	42.6	
Puerto Gaitán (Meta)	1	207.0	1.09	1	NA	0.20	NA	0.86	Inf	
San Martín (Meta)	2	297.3	2.36	0	1.67	0.41	0.78	0.42	Inf	
Notes.

Bolded values indicate significance.

Populations cluster analysis

Analysis of Bayesian Information Criterion (BIC) for increasing numbers K of discriminate analysis of principal components (DAPC) clusters indicated greatest support for three clusters within the dataset (Fig. S2). This inference was supported by the results of STRUCTURE analysis, with delta K most strongly supporting presence of three clusters among the sampling sites (Fig. S3). The clusters were partitioned geographically (Fig. 2). Population One encompassed the Mexico samples provided by A. Piaggio (United States Department of Agriculture) used as a comparator. Two populations were distributed across Colombia, with most of Population Two occurring in high-elevation portions of Colombia, east of the central Andes Mountain passage (Figs. 2 & 3) (807.5 m average elevation of capture). Population Three, in contrast, was distributed across central and western portions of the study area, slightly lower in elevation (Fig. 2) (575.2 m average elevation of capture). One sampling site (Agua de Dios) with the largest sampling effort (40 individuals) was split between populations two and three. We used the three clusters identified from the DAPC analysis as hierarchical strata (i.e., populations), while the respective sampling sites were considered subpopulations to calculate descriptive genotypic metrics (e.g., allelic variation, heterozygosity, and genetic distances) across both sampling sites and inferred populations. Observed heterozygosity (Ho) was less than expected heterozygosity (He) for all sampling sites and populations (Table 1).

Figure 2 DAPC cluster distribution of genotype data by collection.

(A) Visual representation of each population cluster identified by our discriminant analysis of principal components (DAPC). The axes represent the first two principal components with two discriminant analysis eigenvalues (DA eigenvalues, inset) summarizing 75.0% of the data. (B) Geographic distribution of DAPC populations. The colors of the respective DAPC clusters and geographic locations are the same. The first population consists of only six individuals, which was utilized as a comparator for this study. Maps created using ArcGIS Pro software version 2.5 with shape files from DIVA-GIS (Hijmans, 2017; Esri, Redlands, CA, USA, 2019).

Figure 3 Topographic relief and population distribution.

Location of sampling sites across the Andes Mountain range. Inset map shows the central Andean corridor, where higher rates of migration between sampling sites were detected. Partition of DAPC populations Two and Three (inset map on right) are shown as well. Observe the topographic pattern in the partitioning of populations from Population Three to Population Two. Maps created using ArcGIS Pro software version 2.5 with shape files from DIVA-GIS (Hijmans, 2017; Esri, Redlands, CA, USA, 2019).

Genetic variation and migration between and among populations

Results of AMOVA analysis showed significant genetic variance (27.7%, p = 0.01) within the respective sampling sites (Table 2). We also identified significant variance between sampling sites within populations (5%, p = 0.01), and between populations in Colombia (14.9%, p = 0.01) (Table 2). Populations Two and Three from Colombia had similar mean numbers of alleles per locus (7.4 and 7.1, respectively) and allelic richness (4.9 and 4.5) (Table 1). The inbreeding coefficient FIS for all populations was between 0.31 and 0.33, indicating an excess of homozygotes above Hardy-Weinberg equilibrium expectations (Campagne et al., 2012), suggesting inbreeding or family structure. Estimated effective population sizes for the populations in Colombia were 16.9 for Population Two and 41.6 for Population Three, but were unbounded (Table 1). High genetic distances (FST∼0.30, G’ST∼0.80) were observed between Population One in Mexico and Populations Two and Three in Colombia (Table 3), as was expected since Population One was at a greater distance from the others, had a small sample size, and was utilized as a comparator. Genetic distances between Populations Two and Three in Colombia were comparatively low (FST = 0.12, G’ST = 0.34). The estimated equilibrium rate of migration across an ecological time-scale (i.e., genetically effective migration) between Populations Two and Three was 1.83 individuals per generation. Estimates of genetically effective migration between sites indicated that there was historic migration (>1 individuals per generation) between sites that were closer together (<120 km) and located on either side of the Río Magdalena valley, Llanos Orientales, and Orinoquia of the central Andean corridor (i.e., Ibague and Chaparral, Tolima, Yopal, Casanare, Agua de Dios, Cundinamarca, and San Martín, Meta) (Table S2). GeneClass2 direct migrant assessment revealed 15 likely first-generation migrants. Ten of these likely migrants were captured in Population Three (Table S3) and the remaining five in Population Two. Migrants were captured at sites Agua de Dios and Puente Quetame in Cundinamarca, Coello and Piedras in Tolima, Yopal in Casanare, and at San Martín and Pipiral in Meta. Identifying first-generation migrants between local collection sites was not possible due to limited sample size from some such sites. FSTacross all females sampled for this study was higher (0.24) than for all males sampled (0.14). Only females at the lowest distance class (60.4) calculated for our samples demonstrated significant positive spatial autocorrelation (r = 0.31, p = 0.001) (Fig. 4). Males had positive spatial autocorrelation at larger distance classes (438.8, r = 0.06), but this was not significant per our 95% confidence interval (p = 0.08). Females had stronger fine-scale genetic structure than males at short geographic distances. Assessment of differential dispersal by sex for each independent population was not possible due to limited power for Population Three, where only five females were captured.

Table 2 Analysis of molecular variance (AMOVA) between and among collection of D. rotundus populations and sampling sites analyzed.

Variance and significance values within and between both Colombia populations delineated by DAPC and subpopulation (i.e., sample site locations). Significant variance was present across all strata tested (i.e., p values < 0.05). Most covariance originated from within the samples.

Source	Degrees of freedom	Sum of squares	Mean square	Sigma	% Covariance	ϕ	p value	
Between populations	2	164.93	82.47	1.28	14.89	0.15	0.01	
Between sample sites within populations	20	239.34	11.97	0.43	5.01	0.06	0.01	
Between individuals within each sample site	69	640.05	9.28	2.38	27.70	0.35	0.01	
Within individuals	92	414.81	4.51	4.51	52.40	0.48	0.01	
Total	183	1,459.14	7.97	8.61	100	–	–	

Table 3 Metrics of genetic distance between populations.

FST (below diagonal) and G’ST (above diagonal) metrics of genetic distance between populations identified by DAPC. Pairwise FST (Yang, 1998; Hedrick, 2005) was assessed using dartRverse. Greater genetic distance is signified by large numbers. The largest genetic distance was present between populations one and three. Less genetic distance was present between Colombian populations two and three, indicating greater gene flow between the two populations.

	Population 1	Population 2	Population 3	
Population 1	–	0.78	0.83	
Population 2	0.28	–	0.34	
Population 3	0.31	0.12	–	

Figure 4 Genetic structure across distance by sex.

To assess whether fine-scale genetic structure was evident based on sex, we conducted a one-tailed permutation test to identify autocorrelation coefficients (r). We then estimated 95% bootstrap confidence intervals (whiskers around points) around the value for r for each sex at different distances classes based on our samples. X axis distance classes represent geographic distance classes (Euclidean distance rather than simple distance) calculated across standardized across our samples. In this case, if the bootstrap confidence intervals do not overlap zero, then fine-scale genetic structure is present. Only female individuals at the lowest distance class showed significant positive spatial autocorrelation (r = 0.37, p = 0.001).

Environmental differences between and among populations

We confirmed that the environmental variables we chose to assess at the locations of capture for our samples varied with the elevational gradient used for sample site selection via a multiple linear regression model (F = 2267, p < 0.001). Multiple t-tests of environmental characteristics at the respective sampling locations showed no significant differences between the environments of each population in Colombia (all p > 0.05). The MANOVA of environmental variables did not elucidate any significant difference between populations (Pillai’s Trace = 0.08, F (9,70) = 0.65, p = 0.75). Population Three occurred at an overall lower average elevation (mean elevation 575.2 m) than Population Two (mean of 807.5 m), a difference which was not significant statistically (t = 1.91, df = 71.72, p = 0.06). We identified a positive but non-significant relationship between genetic distance (FST, G′ST, or Euclidean distance) and geographic distance (km) (Mantel statistic = 0.05, p = 0.27, Mantel statistic = 0.07, p = 0.18, and Mantel statistic = 0.05, p = 0.23, respectively). No significant relationship was present between the elevation of sampling sites and genetic distance (Mantel statistic = 0.05, p = 0.15).

Discussion

This study’s identification of two genetically distinct but interconnected D. rotundus populations in Colombia provides a significant insight into the potential mechanisms of RABV spread. The relatively low genetic differentiation between these populations based on both the FST (0.12) and G′ST (0.34) metrics (Table 3), despite considerable geographic distances (21.6–696.7 km) supports the inference of both long-term migration and recent dispersal, potentially along low-elevation corridors such as the Rio Magdalena valley. The presence of first-generation migrants, especially individuals originating in the east but captured in the west, implies that current movement patterns may mirror historical migration, sustaining connectivity across the region. The presence of two populations at one sampling location (Agua de Dios), which rests at the center of the Central Andean corridor, supports this interpretation.

Given that D. rotundus dispersal is known to be limited to relatively short range (10–20 km) (Trajano, 1996; Streicker et al., 2012b; Rocha & Dias, 2020), and short-range dispersal has been related to slow wavefronts of outbreaks of D. rotundus- associated RABV (Benavides, Valderrama & Streicker, 2016), the observed genetic connectivity suggests that even moderate dispersal rates may enable RABV to cross regional ecological and geographical barriers. This dispersal is particularly concerning in areas where livestock densities are high, as infected bats could transmit RABV across populations despite physical separation by elevation or landscape. Overall, the consistency of this dispersal with previous findings and the importance of low-resistance corridors strengthen the ecological inference that migration and, potentially, viral flow in D. rotundus is governed more by geographic permeability than strict distance or habitat type (Pinto, 2009; Das et al., 2019; Seetahal et al., 2024)

Allelic richness, a metric of genetic diversity (Foulley & Ollivier, 2006), also showed similar genetic diversity within the two Colombian populations (4.91 and 4.49) (Table 1). The small number of populations identified, coupled with the limited genetic distance between these two populations indicates that D. rotundus colonies in Colombia are interconnected by migration (Martins et al., 2009; Streicker et al., 2012a; Streicker et al., 2016). Other research on genetic structure of D. rotundus in French Guiana using microsatellites and mitochondrial DNA found a greater degree of historic migration at a similar geographic scale to that in our study (i.e., regional) when compared to larger-scale comparisons between North and South America (Huguin et al., 2018). The spatial scale and geographic locations of previous studies may account for any differences between our results and those for studies done in different locations.

Our estimate of the equilibrium rate of migration (m = 1.83) showed long-term connection between Colombian populations, and our direct migrant identification assessment identified many first-generation migrants within the sampled generation (n = 15 of 86 individuals assessed). Most first-generation migrants (n = 10) originated in the eastern population in Colombia (Population Two), but were captured in the western population (Population Three) (Fig. 4). All migrants were captured at lower elevations (<1,000 m). The demonstration of interconnected bat populations across wide regions despite presumed physical barriers (e.g., the Andes Mountains) implies that regional rabies control cannot be confined to localized hotspots. Surveillance systems should be designed to track RABV not just where outbreaks occur, but in regions of potential bat movement, especially along valleys and lowland corridors.

No significant differences in environment or climate of occupancy were apparent between the two inferred populations, nor were there differences in anthropogenic factors such as agricultural activity (i.e., cattle density) or human population density (p > 0.05). Greater genetically effective migration rates were inferred between sites surrounding the Río Magdalena valley of the central Andean corridor, with sites to the east (Population 2) having the most apparent migration (Fig. 4, Table S2). The two inferred populations were geographically separated by the Andes Mountains to the east of the valley (Fig. 4). Other studies of D. rotundus-associated RABV have found that this bat may disperse through low-resistance areas such as valleys (Benavides, Valderrama & Streicker, 2016; Streicker & Allgeier, 2016), a finding convergent with our results. As such, our results support the potential role of elevation or topography in shaping D. rotundus dispersal, and hence we suggest that livestock vaccination campaigns prioritize areas adjacent to high-migration zones, such as the Río Magdalena valley.

Previous research using mitochondrial DNA also has found low genetic divergence among D. rotundus populations over large distances in some regions (i.e., Central America and the Pantanal region of Bolivia), while other regions in the same study had large genetic divergence over short distances (i.e., Atlantic forests in southern Brazil) (Martins et al., 2009). The authors hypothesized that greater genetic divergence may be driven by differences in ecoregion, rather than by distance per se for this species due to its volant nature. These results are consistent with those of our study, as distance among sites did not relate significantly to genetic differentiation. Nevertheless, other studies using the same microsatellite loci as this study have found relatively low genetic differentiation across different ecoregions for D. rotundus at the leading edge of the species’ broader geographic range in Mexico (Piaggio et al., 2017). In contrast, another study in Mexico using methods similar to those of our study found a higher degree of genetic diversity across a larger sample size and geographic area (Romero-Nava, León-Paniagua & Ortega, 2014). The lack of consensus among previous studies, and the lack of significant geographic or landscape patterns between populations in this study, reflects variation of geographic population genetic patterning in D. rotundus across its range. Results from this study may support previous hypotheses of male-biased dispersal for D. rotundus (Martins et al., 2009; Streicker & Allgeier, 2016). Females showed positive spatial autocorrelation as lower distance-classes than males, and positive spatial autocorrelation for males was not significant even at greater distances (Fig. 4). Our finding of related males potentially being collected at locations much farther apart than females echoes results from previous studies (Streicker & Allgeier, 2016). Nevertheless, previous studies have assessed inbreeding coefficients (FIS) by sex and by population to further assess male-biased dispersal (Streicker & Allgeier, 2016). Assessment of FIS by sex and by population was not possible here, however, due to the limited number of females—just five—collected in Population Three. Dispersal of infected D. rotundus individuals could facilitate RABV transmission between subpopulations (Streicker et al., 2012b; Bakker et al., 2019; Becker et al., 2021; Horta et al., 2022).

A limitation of this study is the lack of RABV prevalence data for the individuals collected. Hence, we are unable to directly assess the effect of genetic structure of D. rotundus in Colombia upon RABV spillover. RABV spillover occurrence data are available from Colombia, and are collected via the Epidemiological Information and Surveillance System of ICA (Instituto Colombiano Agropecuario (ICA), 2019). This organization performs surveillance for RABV outbreaks in livestock and reports suspected cases from local-level stakeholders (Rojas-Sereno et al., 2022; Instituto Colombiano Agropecuario (ICA), 2023). Recent reports have identified a persistent concentration of cases in the Northern Caribbean region of the country, mostly in cattle, and an expansion of RABV cases into previously naive areas (Rojas-Sereno et al., 2022). Between 1982 and 2023, Instituto Colombiano Agropecuario (ICA) (2024) recorded 3878 rabies outbreaks in animals (89% in cattle), distributed across 31 of Colombia’s 32 departments. Cases peaked in 2014 (542 cases), but have declined to as few 43 cases reported in 2019 (Bonilla-Aldana et al., 2022). The Caribbean region remains at high risk, however, with constant impact in the departments Cesar, Magdalena, Bolívar, and Sucre (Bonilla-Aldana et al., 2022). In the Orinoquia region (east of the Andes Mountains), 13 spatial clusters were identified in Vichada, in addition to cases in Casanare and Meta (Bonilla-Aldana et al., 2022). During the 2005–2019 period, the departments with the most reported cases of RABV were Cesar (18.6%), Magdalena (11.1%), Arauca (9.3%), Casanare (8.5%), and Sucre (6.6%) (Bonilla-Aldana et al., 2022). In 2020, most bovine rabies outbreaks were concentrated in Sucre, indicating focalized viral circulation that requires continuous epidemiological monitoring (Instituto Nacional de Salud, INS). Our findings of broader interconnection between D. rotundus populations in Colombia across many of these regions could help explain how RABV is spread across the country, particularly from eastern locations with higher cattle density to other, less agriculturally dense areas (Rojas-Sereno et al., 2022). To build upon the assessment conducted in this study, future assessments with a broader sampling base, potentially in the Caribbean region where cases are concentrated, could connect D. rotundus populations genetics information with RABV spillover data and coupled antibody tests for field samples.

Our sample size was smaller than those of previous assessments (Romero-Nava, León-Paniagua & Ortega, 2014; Piaggio et al., 2017), though our samples were collected over a narrower time-range. Assessment of a greater sample size across a broader geographic area or more diverse environments may further elucidate the impacts of environmental variation upon population genetic patterning of D. rotundus in Colombia. Utilizing only amplified fragment-size analysis for scoring microsatellite markers may have limited our observation of genetic variation as well. Other methods, such as microsatellite genotyping by sequencing (SSR-GBS) or whole-genome sequencing for single nucleotide polymorphisms (SNPs), could have shown more detailed patterns of genetic structure (Tibihika et al., 2019). Future research in this system should work to increase the geographic scope of sampling to capture more ecoregions or fine-scale landscape factors that may shape genetic differentiation for D. rotundus populations. Future research should also couple genetic assessments with RABV seroprevalence assessment to more directly test hypotheses of RABV transmission driven by dispersal of infected D. rotundus individuals.

Conclusions

This study aimed to establish a baseline assessment of population genetic structure of D. rotundus individuals across an elevational gradient in Colombia, with the broader goal of evaluating how genetic connectivity may influence RABV transmission dynamics. Screening 11 microsatellite loci developed by Piaggio, Johnson & Perkins (2008) and applying standard population genetic analyses, we quantified genetic variation and assessed the geographic distribution of genetic clusters. Comparative analysis with a reference population from Mexico allowed us to contextualize our findings within a broader regional framework. Our results revealed substantial genetic connectivity among Colombian D. rotundus populations, even across geographically complex terrains. This finding supports the hypothesis that interpopulation bat movement contributes to the dissemination of RABV. While environmental and anthropogenic barriers showed limited impact, natural geographical features such as valleys appeared to provide important movement corridors. These results indicate that it is difficult to predict vampire bats’ gene flow and dispersal, especially within the context of predicting RABV transmission between populations, as has been noted in previous research (Streicker et al., 2012b; Blackwood et al., 2013).

Our results contribute to current knowledge of the regional distribution of genetic variation for D. rotundus. Our results demonstrate that D. rotundus populations in Colombia exhibit significant genetic connectivity, even across complex landscapes, supporting the hypothesis that interpopulation bat movement can facilitate RABV spread. These findings advance our understanding of the spatial dynamics of vampire bat populations and underscore the relevance of incorporating ecological and genetic data into public health planning. Effective rabies control strategies should consider the role of landscape connectivity in shaping bat movement and virus spread. A more detailed assessment of the role of elevation is warranted to fill remaining knowledge gaps; specifically, future studies should incorporate denser sampling along elevational gradients, finer-scale genotyping (e.g., SNP-based approaches), and integration of environmental niche modeling to better understand how elevation and landscape features influence population structure and viral transmission risk. While environmental and anthropogenic differences between sites occupied by the respective populations were minimal, geographical corridors such as valleys play a key role in movement. Such integrative approaches can support more ecologically informed public health policies and practices, including targeted vaccination, strategic land-use management, and interdisciplinary surveillance.

Supplemental Information

Supplemental Information 1 Null Allele Frequency in Desmodus rotundus samples from Colombia

Estimated frequencies of null alleles at each of the 12 microsatellite loci for Desmodus rotundus. Null allele frequency was estimated using the Brookfield (1996) method in MicroChecker (Van Oosterhout et al., 2004) . Loci with null allele frequency >0.35 were eliminated from consideration (i.e., locus Dero_H02).

Supplemental Information 2 DAPC Cluster analysis of Desmodus rotundus genotype data by collection

Bayesian information criterion (BIC) model selection criterion for progressive numbers of DAPC clusters (K) for identification of the best-supported number of clusters within our samples. We tested one to 15 clusters (three more than the number of sampling sites). Results indicate that three clusters of genetically similar individuals (i.e. populations) are most likely present within the data.

Supplemental Information 3 STRUCTURE-based cluster analysis of Desmodus rotundus microsatellite data

Analysis of STRUCTURE-based clusters (K) (Pritchard, Stephens & Donnelly, 2000) conducted using the delta K method of Evanno et al. (2005). A) Mean likelihood of each cluster from 20 replicate runs for each value of K. B) Delta K (mean absolute value of average likelihood over 20 runs divided by the standard deviation of mean likelihood) of each progressive number of clusters. Analysis of delta K was performed using STRUCTURE version 2.3.4 (Pritchard, Stephens & Donnelly, 2000).

Supplemental Information 4 Sample sites for collection of tissue from Desmodus rotundus individuals

Sampling sites (i.e. locations of collection) were located across an elevational gradient from low (<500 meters in elevation), to moderate (500–1,000 meters in elevation) to high (>1,000 meters in elevation). This elevational gradient was used as a proxy for environmental variation through which we assessed multiple factors as they relate to the structure of population genetics of D. rotundus in Colombia after analysis. Samples for this study were collected in the summers of 2022 and 2023. Five samples were provided by the Instituto de Investigación de Recursos Biológicos Alexander von Humboldt (IAvH) (Bogotá, Colombia) and were collected in Colombia between 2019 and 2022. As a comparator, we amplified DNA from six D. rotundus individuals collected in western Mexico in 2007 by Piaggio, Johnson & Perkins (2008).

Supplemental Information 5 Metrics of genetic distance between sampling sites

FST and GST′ metrics of genetic distance between each sampling site of D. rotundus individuals. Greater genetic distance is signified by large numbers. Sampling sites Nuevo Leon and Tamaulipas were located in Mexico and were used as a comparator for all other sites located in Colombia.

Supplemental Information 6 Effective Migration Rate (m) between sampling sites

Metrics of effective migration between collection sites in Colombia using both Wright’s FST equation for effective migration (m) (Wright, 1951; Zhivotovsky, 2015). Effective migration or “equilibrium” migration rates can be interpreted as individuals migrants per generation across ecological time calculated using FST, which is a summary statistic of genetic differentiation (Wang & Whitlock, 2023; Yamamichi & Innan, 2012). High migration rates (>1 individual per generation) was present between sites surrounding the Río Magdalena valley.

Supplemental Information 7 First-generation migration probability between populations

Migrant identification was conducted in GeneClass 2.0 (Piry et al., 2004) . This software identifies likely first-generation migrants (shown in red) and the likelihood that these migrants originated from other sampled populations (Piry et al., 2004) . The lowest metrics of -log(L) indicate the most likely population of origin (shown in green). 15 first-generation migrants were identified.

The authors thank our collaborators from Colombia, including Natalia Cediel, Gelys Igreth Mestre Carrillo, Fernando Sarmiento Parra, Catalina Cárdenas González, Catherine Mora, Hugo Fernando López Arévalo, Andrew Jackson Crawford, Sebastián García Restrepo, Sandra Patricia Galeano Muñoz, Andrés Julián Lozano Flórez, Elizabeth Aya Baquero, Francisco Alejandro Sánchez Barrera, Giovany Guevara Cardona, Gladys Reinoso Flórez, Leidy Azucena Ramírez Francel, Leidy Viviana García Herrera, Hector Ramírez Chaves, Andrea Bustamante Cadavid, Victor Hugo Serrano Cardozo, Alexander Velásquez Valencia, Oscar Marín Ducuara, and Alejandro Rocha. Special thanks to the students and technicians who collaborated in the data collection, including Mariana Castaneda Guzman, Quan Dong, Shariful Islam, Julia Alexander, Laura Valentina Ávila Vargas, Nicolas David Sanabria Rivera, Karen Sarmiento, Juan Camilo Quintero Navarro, Carlos Dubon Hinojosa, Kaitlyn Enstice, Carlos Bravo, and Alirio Rey (Don Ali). They also acknowledge Antoinette Piaggio and Annie Tibbels for the provision of bat samples from Mexico for the completion of this study.

Additional Information and Declarations

Competing Interests

Author Contributions

Animal Ethics

Field Study Permissions

Data Availability

The authors declare there are no competing interests.

Paige Van de Vuurst conceived and designed the experiments, performed the experiments, analyzed the data, prepared figures and/or tables, authored or reviewed drafts of the article, and approved the final draft.

Analorena Cifuentes-Rincon conceived and designed the experiments, performed the experiments, authored or reviewed drafts of the article, and approved the final draft.

Andrea S. Bertke conceived and designed the experiments, analyzed the data, authored or reviewed drafts of the article, and approved the final draft.

Diego Soler-Tovar conceived and designed the experiments, authored or reviewed drafts of the article, and approved the final draft.

Nicolás Reyes-Amaya performed the experiments, authored or reviewed drafts of the article, and approved the final draft.

Fabiola Rodriguez Arévalo performed the experiments, authored or reviewed drafts of the article, and approved the final draft.

Julieth Stella Cárdenas Hincapié performed the experiments, authored or reviewed drafts of the article, and approved the final draft.

Jhon Rivera-Monroy performed the experiments, authored or reviewed drafts of the article, and approved the final draft.

Luis E. Escobar conceived and designed the experiments, analyzed the data, authored or reviewed drafts of the article, and approved the final draft.

Eric Hallerman conceived and designed the experiments, analyzed the data, prepared figures and/or tables, authored or reviewed drafts of the article, and approved the final draft.

The following information was supplied relating to ethical approvals (i.e., approving body and any reference numbers):

Capture and handling protocols followed recognized bat handling guidelines of the American Society of Mammalogists Sikes (2016), as approved by the Institutional Animal Care and Use Committee at Virginia Tech (IACUC approval #21-138). Approval was also obtained from the Institutional Care and Use Committee of the Universidad de La Salle (IACUC approval #087).

The following information was supplied relating to field study approvals (i.e., approving body and any reference numbers):

Capture of bats and data collection in Colombia were carried out under environmental permit Resolución 1473 de 2014 of the Autoridad Nacional de Licencias Ambientales (ANLA) (i.e., UniSalle Wild Species Collection Framework Permit).

The following information was supplied regarding data availability:

The data is available in the Supplemental Files.

The raw data and code are available at Figshare:

- https://figshare.com/projects/Raw_Data_and_Code_for_Assessing_the_effects_of_environmental_variation_on_population_genetic_structure_of_the_common_vampire_bat_in_Colombia_/248969. - Van de Vuurst, Paige (2025). Sample data and meta data for analyses. figshare. Dataset. https://doi.org/10.6084/m9.figshare.29078804.v1

The data for locus: Dero_B10F_E01R. is available at Figshare: Van de Vuurst, Paige (2025). Raw data for locus: Dero_B10F_E01R. figshare. Dataset. https://doi.org/10.6084/m9.figshare.29078033.v1

The data for locus: Dero_B03F_B03R is available at Figshare: Van de Vuurst, Paige (2025). Raw data for locus: Dero_B03F_B03R. figshare. Dataset. https://doi.org/10.6084/m9.figshare.29077775.v1

The data for locus: Dero_B11F_B11R is available at Figshare: Van de Vuurst, Paige (2025). Raw data for locus: Dero_B11F_B11R. figshare. Dataset. https://doi.org/10.6084/m9.figshare.29078144.v1

The data for locus: Dero_C12F_B02R is available at Figshare: Van de Vuurst, Paige (2025). Raw data for locus: Dero_C12F_B02R. figshare. Dataset. https://doi.org/10.6084/m9.figshare.29078294.v1

The data for locus: Dero_D06F_D06R is available at Figshare: Van de Vuurst, Paige (2025). Raw data for locus: Dero_D06F_D06R. figshare. Dataset. https://doi.org/10.6084/m9.figshare.29078336.v1

The data for ocus: Dero_C07F_A02R F is available at Figshare: Van de Vuurst, Paige (2025). Raw data for locus: Dero_C07F_A02R F. figshare. Dataset. https://doi.org/10.6084/m9.figshare.29078345.v1

The data for locus: Dero_D12F_D12R is available at Figshare: Van de Vuurst, Paige (2025). Raw data for locus: Dero_D12F_D12R. figshare. Dataset. https://doi.org/10.6084/m9.figshare.29078441.v1

The data for locus: Dero_G10F_B03R is available at Figshare: Van de Vuurst, Paige (2025). Raw data for locus: Dero_G10F_B03R. figshare. Dataset. https://doi.org/10.6084/m9.figshare.29078492.v1

The data for locus: Dero_H02F_C03R is available at Figshare: Van de Vuurst, Paige (2025). Raw data for locus: Dero_H02F_C03R. figshare. Dataset. https://doi.org/10.6084/m9.figshare.29078504.v1

The data for locus: Dero_A08F_B01R is available at Figshare: Van de Vuurst, Paige (2025). Raw data for locus: Dero_A08F_B01R. figshare. Dataset. https://doi.org/10.6084/m9.figshare.29078651.v1

The data for locus: Dero_C11F_C11R is available at Figshare: Van de Vuurst, Paige (2025). Raw data for locus: Dero_C11F_C11R. figshare. Dataset. https://doi.org/10.6084/m9.figshare.29078699.v1

The data for locus: Dero_D02F_D02R is available at Figshare: Van de Vuurst, Paige (2025). Raw data for locus: Dero_D02F_D02R. figshare. Dataset. https://doi.org/10.6084/m9.figshare.29078738.v1

The data for Environmental Values at location of capture is available at Figshare: Van de Vuurst, Paige (2025). Environmental Values at location of capture for D. rotundus] in Colombia. figshare. Dataset. https://doi.org/10.6084/m9.figshare.29583239.v1

The sample data and meta data for analyses is available at Figshare: Van de Vuurst, Paige (2025). Sample data and meta data for analyses. figshare. Dataset. https://doi.org/10.6084/m9.figshare.29078804.v1

The code for Population Genetics Assessments is available at Figshare: Van de Vuurst, Paige (2025). Code for Population Genetics Assessments. figshare. Online resource. https://doi.org/10.6084/m9.figshare.29077340.v1.

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
