# Peer review of "A preliminary assessment of population genetic structure of the common vampire bat (Desmodus rotundus) in Colombia"

_PeerJ, doi:10.7717/peerj.20306_

## Round 0.1 · original submission · Major Revisions

· Academic Editor

Major Revisions

All reviewers feel that the manuscript requires major revisions to become suitable for publication. Please consider, and respond to, all of the feedback from the reviewers in your revisions.

**PeerJ Staff Note**: Please ensure that all review, editorial, and staff comments are addressed in a response letter and that any edits or clarifications mentioned in the letter are also inserted into the revised manuscript where appropriate.

**PeerJ Staff Note**: It is PeerJ policy that additional references suggested during the peer-review process should only be included if the authors agree that they are relevant and useful.

**Language Note**: The review process has identified that the English language must be improved. PeerJ can provide language editing services - please contact us at [email protected] for pricing (be sure to provide your manuscript number and title). Alternatively, you should make your own arrangements to improve the language quality and provide details in your response letter. – PeerJ Staff

Reviewer 1 ·

Basic reporting

Title: The current Title does not specify the objectives or indicate the prime finding of the study.
Consider the following suggestion to revise the Title: “Landscape heterogeneity has minimal influence on the population genetic structure of the common vampire bat (Desmodus rotundus) across elevational gradients in Colombia.”

Abstract: The unstructured Abstract is concise but does not justify the study and lacks key information. Consider the following suggestion to enhance the significance of the findings.
• The relevance of the study is unclear. Please justify the need for the study in this area.
• Provide statistical significance values for comparative analysis data.
• Line 47: “Our results support previous hypotheses of male-biased and resistance-mediated patterns of dispersal for D. rotundus” – The sentence is not supported by any data. Consider revising or providing data in the abstract.
• If possible, provide adequate reasoning as to why no correlation was observed between the landscape heterogeneity and the population genetic structure of D. rotundus. Also, indicate the potential implications of these findings.

Introduction
The Introduction is well-written and establishes a link between landscape degradation, wildlife health, and the emergence of zoonotic disease. However, consider incorporating the following information to enhance the significance of the study.
Suggestions:
• The role of landscape degradation has been established in zoonotic diseases; however, the manuscript does not specify the need for additional studies in this area. Please justify the need for the present study. Also, clarify how the knowledge generated in this study might differ from the findings highlighted in “Streicker DG, Recuenco S, Valderrama W, et al. Ecological and anthropogenic drivers of rabies exposure in vampire bats: implications for transmission and control. Proc Biol Sci. 2012;279(1742):3384-3392. doi:10.1098/rspb.2012.0538.”
• In the current format, the manuscript does not report a correlation between landscape heterogeneity and population genetic structure. Hence, it is crucial to enlist the other factors that could play a role in the emergence of zoonotic disease.
• Elaborate on other studies that have established a strong correlation between the population genetic structure of wildlife and zoonotic diseases.
• Justify why elevation was considered a priority parameter among the other environmental factors that might affect genetic diversity.
• Describe the current strategies for surveillance of RABV.
• Include a statement of whether the current findings could be effective in conducting surveillance of zoonotic diseases and formulation of response strategies.
Consider adding the suggested references to fill in the information gaps and support the above claims:
1. Streicker DG, Recuenco S, Valderrama W, et al. Ecological and anthropogenic drivers of rabies exposure in vampire bats: implications for transmission and control. Proc Biol Sci. 2012;279(1742):3384-3392. doi:10.1098/rspb.2012.0538
2. Biek R, Real LA. The landscape genetics of infectious disease emergence and spread. Mol Ecol. 2010;19(17):3515-3531. doi:10.1111/j.1365-294X.2010.04679.x

Figures & Tables
The Figures and Tables are adequate. However, add a footnote to the Tables and a caption to the Figure Legends indicating the statistical tools employed for the analysis of each piece of data. Also, indicate the level of statistical significance.

Experimental design

Materials and Methods
The Materials and Methods section is structured adequately. However, additional information is requested to enhance the reproducibility of the data presented in the manuscript.
• Line 111: “86 tissue samples from different individuals.” – The genetic diversity can be influenced by sex and age. Therefore, expand on the demographic information.
• Indicate which inclusion or exclusion criteria were followed during sampling.
• Indicate whether the sampled bats were in healthy or diseased conditions.
• Line 141: “Both capture and release sampling as well as lethal sampling were…” – Please justify why lethal sampling was conducted when capture and release was potentially available.
• Indicate how the quality of the isolated DNA was ascertained.
• Add the raw data to an applicable repository.
• Provide information on the statistical analysis for AMOVA and Hardy-Weinberg tests.
• Justify why multiple t-tests were used instead of multivariate analysis.

Validity of the findings

Results
Although the Results have been presented appropriately, please include the following information to fill in the gaps in the findings based on the data.
• Lines 267-268: Justify why the small size of the comparator group does not undermine the robustness of the estimates of genetic distance.
• For clear understanding, consider structuring the section with distinctive subheadings to distinguish the specific findings.
• The data can be interpreted by relying heavily on the statistical analysis. Therefore, justify each of the statistical tools employed and describe the statistical significance values in detail in the Methods section.

Discussion
Please incorporate the following information to fill in the gaps in the Discussion section and correlate the findings with the literature.
• Line 317: “Our assessment also confirmed male-biased dispersal for D. rotundus individuals…” – As indicated in line 303, only five females were captured, indicating that the sample size is too small to derive statistical significance. Therefore, this sentence can be removed as the findings are not supported by adequate data.
• Line 318: “These trends, ….” – As the Conclusion is not supported by statistical data, further research might not be essential. Hence, the authors could make a suggestive statement to avoid negation of future investigations.
• Discuss how the findings presented in the study could be factors in the spread of RAVB.
• Provide a comparative analysis of the current findings with the literature on the method of sampling, statistical tools, and environmental factors. Together, these parameters can provide different interpretations of data.
• Discuss how the findings could assist in developing policies on rabies control and its influence on public health policies.

Conclusion
The Conclusion is primarily written as future directions. Please include a first paragraph to summarize the main findings of the study. Also, the prevalence of RABV was estimated within the sampled bats. Therefore, a direct link between the RABV reservoir, transmission, and landscape variability cannot be established. Therefore, the conclusion statement needs to be speculative rather than confirmatory.

·

Basic reporting

Thanks for the opportunity to review this manuscript. The authors address a relevant question—how landscape heterogeneity shapes population structure in Desmodus rotundus, with implications for rabies virus dynamics. The use of microsatellite markers across an elevational gradient in Colombia is a compelling approach, and the study is well situated to contribute to our understanding of bat movement ecology and disease transmission. That said, the introduction could be significantly improved for clarity and focus. It is currently too long, with several sections repeating ideas or introducing tangents that are not clearly tied to the study's main objective, which is only clearly articulated in the final paragraph. The earlier discussion of zoonotic spillover and immune suppression could be shortened and more directly linked to the ecological drivers of gene flow and RABV spread in D. rotundus. The parts of the introduction that are most relevant—rabies ecology, dispersal, and genetic structure—should be brought forward to better frame the research question. Some points also require correction: RABV is not strictly tropical, and while bats are the primary reservoir in Latin America, this isn’t true for other tropical regions. Furthermore, the sentence suggesting that RABV transmission is a function of gene flow is misleading—dispersal may influence transmission, but gene flow implies reproduction and shouldn’t be conflated with movement. Additionally, “pathogen spillover transmission” is redundant and could be replaced with “pathogen spillover” or “cross-species transmission.” Finally, the Plowright references do not support the claim that wildlife mortality in disturbed environments leads to decreased genetic diversity, and more appropriate sources should be cited if this point is retained.

Regarding tables and figures, the manuscript would benefit from clearer and more informative visual representations of the sampling design. Rather than Table 1, a map showing the geographic locations of sampling sites overlaid with elevation gradients would provide better context for the spatial and environmental analyses. Perhaps the map should also include information on agricultural activity or human population density to aid in the interpretation of the data. Currently, Figure 1 does not include elevation—despite its relevance for both sampling design and interpretation of genetic structure—and lacks a scale bar. Figure 2B is more informative for visualizing population structure, and Figure 4 conveys key spatial and genetic patterns. I suggest combining elements of these figures and redesigning Figure 1 to serve as a comprehensive spatial overview.

Experimental design

Regarding their methods, the authors employ a comprehensive set of analytical tools to evaluate population structure, diversity, and connectivity in D. rotundus, combining both model-based (STRUCTURE) and model-free (DAPC) clustering methods, AMOVA, diversity metrics, and migration estimates. This approach is well-suited to the research questions and provides a strong framework for understanding genetic structure across the landscape. However, I recommend greater clarity in how each analysis is conceptually linked to the study’s objectives—right now, the methods read more like a list of tools than a cohesive analytical workflow. Also, the rationale for some choices needs further justification. For example, the removal of loci with null allele frequencies >0.5 should be explained—this is a very high threshold, and loci with such high frequencies are typically excluded much earlier due to concerns over genotyping error. It would be helpful to report how many loci were removed and assess whether this influenced diversity or structure estimates. I also suggest caution in interpreting Ne estimates, which are sensitive to sampling design and temporal assumptions, and note that the use of multiple t-tests to examine environmental associations increases the risk of Type I error; a multivariate approach may be more appropriate here. Overall, the methods are technically sound but would benefit from improved structure, clearer links to the hypotheses, and some refinements in statistical interpretation.
Also, in the methods, the sentence “Bats were then anesthetized using isoflurane while morphometrics, demographic data, and tissue were collected using non-invasive or minimally invasive tools” raises ethical and practical concerns. Using isoflurane anesthesia for basic morphometrics and tissue sampling (e.g., wing punches) seems excessive and is not standard practice in bat field studies unless there are clear animal welfare or handling constraints. If anesthesia was used for all individuals, this should be justified explicitly, including ethical approval details and potential implications for fieldwork and bat welfare.

Validity of the findings

The discussion presents a clear summary of the results and how they relate to previous work, but there are several areas where it could be strengthened. First, the authors acknowledge that the lack of RABV prevalence data from sampled individuals limits their ability to connect genetic structure to spillover risk. While this is valid, I wonder if they could leverage existing data on rabies incidence in Colombia (e.g., in livestock, humans, or other wildlife) to contextualize their findings. Even if direct linkage isn’t possible, acknowledging this potential source of information could strengthen the relevance of their results for public health and disease forecasting.

Relatedly, the paper repeatedly emphasizes the importance of understanding D. rotundus dispersal for rabies control, yet provides little detail on the current rabies situation in Colombia. Expanding briefly on the epidemiological context in the introduction or discussion—especially regarding regional patterns of RABV outbreaks—would help ground the ecological findings in their applied significance.

In the conclusions, the statement that “a more detailed assessment of the role of elevation… is needed to fill knowledge gaps” is too vague. It would be helpful to specify what kinds of data or analyses are needed (e.g., denser sampling along elevational gradients, integration of environmental niche modeling, or finer-scale genotyping). Lastly, the final sentence overreaches and feels generic. I suggest removing or revising it to better reflect the study's specific contributions rather than invoking broad themes of global climate change.

Reviewer 3 ·

Basic reporting

This review was completed by a professor and a PhD student.

Overall, we feel this will be a useful contribution to our understanding of vampire bat population structure. However, we have several points where we request clarification and suggest changes that we think will strengthen the manuscript. Broadly, we have arranged our comments in decreasing order of importance. Most of our comments are in the area of general reporting. The two greatest issues were in the framing, which is largely focused on areas not analyzed or reported by the paper, and the need for additional information to help in assessing the claims of the paper.

Narrative and framing: The most overarching challenge we had with the paper was a strong disconnect between the actual research conducted and reported versus the title, abstract, and framing. The abstract and introduction focus heavily on rabies, disease ecology, spillover, and landscape heterogeneity (e.g., abstract, lines 34 to 41). Additionally, there is a strong emphasis on global change. However, there are no data about these bats as rabies hosts or vectors; the authors found minimal evidence of landscape impacts on population structure; and environmental variation is mentioned in the title, but no environmental or climate data are reported, except that some were tested and did not differ between the populations. We suggest that the authors significantly reframe their narrative to more closely reflect the analyses they did and the results they found.

In line 305, the authors mention that the objective of this study is to conduct a baseline assessment of population genetic structure. We agree that this is the main goal of the study and should be better emphasized in the abstract and introduction. It seems from the overall paper that the authors are adding to a body of literature that demonstrates that vampire bat population structure can be locally variable, emphasizing the importance of assessing the actual landscape in which one is working. In this context, it makes sense for the authors to do this site-specific analysis in an area where they are preparing to assess the potential for rabies spread. The connection to rabies and the environment can be briefly mentioned in the introduction and/or discussion, but this study is too local and focused on neutral population genetic structure to warrant such a focus on these topics.

Need for additional clarification and reporting: There appears to be missing data or areas where data are not represented in the results or figures.

We would like to see the data related to environmental factors, especially when talking about how these data are linked with genetic data, since it is unclear what these data are or the values used. They should be on a table and the maps. It is unclear if this study has enough data to link genetic diversity to environmental factors. For example, by just looking at the two points they have in Colombia over 1000 m (Table 1, line 11), this may not be enough to obtain clear results. Also, the division between high, low, and moderate altitudes seems arbitrary. Would the results change if the authors were to categorize sites simply as high or low, resulting in a more equal number of points? We suggest the authors refer to the literature to determine how others have defined these elevational gradients to justify their choices. Also, of the two populations with different elevations, is there a difference in genetic diversity, significant or not?

How do sampling sites vary in environmental characteristics, not just the populations?

To aid the reader, Figure 1 should indicate the type of landscape, and the sites should be labeled so readers can understand how sites vary in elevation and distance, and to be able to link population genetic data with the locations. Also, in Figures 1 and 2, we suggest the authors add indications of where geographic barriers lie, e.g., where are the Andes? Figure 2 has more information, but it is still unclear. Lines 341 to 358 mention effects of topography and dispersion, lines 400-404 discuss the impact of topography on connectivity; having the barriers indicated on the maps would help the reader understand the conclusions.

Relatedly, in lines 118-121, the authors mention that the elevational gradient was used as a proxy for multiple climatic and landscape factors; did these factors actually correlate with elevation?

The introduction talked about ecosystem degradation, but no descriptions are given of the study area. Which sites are pristine forests, towns, agricultural, or farming sites? This should be in Table 1.

Figure 3 and lines 297-299, 372-378: The text says that females showed spatial autocorrelation up to 165m and males up to 660m. In the graphs in Figure 3, the data for males is not shown up to 165m – why? The figure legend indicates that the confidence intervals overlapping zero indicate a lack of fine-scale genetic structure, which all of them do. So, how did the authors make this conclusion? Why are the confidence intervals not surrounding what I presume are the estimated r values in Figure 3? What is the difference between the whiskers around the points and the shaded areas? Are the shaded areas a null expectation? In Figure 3, it seems the only distance class where the whiskers do not overlap zero is for males at 330m before losing structure again at further distances – what accounts for this? In essence, how did the authors come to the conclusions they did about spatial autocorrelation?

The authors should also provide the raw molecular and climatic data (microsatellite peak calls, etc.).

Other comments and questions:

Can the authors use any of their genetic information (Ne, Ho vs He, etc) to assess the genetic diversity and evolutionary potential of their populations, either within Colombia or in relation to other populations (e.g., French Guiana)?

Disturbed ecosystems or the presence of cows are not necessarily linked to the density of vampire bats in an area or even their presence; therefore, degrading environments are not necessarily linked to rabies, so lines 105 or 106 should be better cited to link the degradation of ecosystems with rabies.

Furthermore, I do not think this study found a correlation between land use and variation of genetic diversity, which is mentioned in lines 121 and 122. These negative results could be interesting to explore since they make it difficult to predict vampire bats' gene flow and mobility habits in general.

Lines 330 to 332 indicate that the metrics of genetic diversity were low. Have you compared them to other species of bats, or are you comparing them to other mammals, or what species? If this statement is true, these results are interesting because they indicate the great mobility of this organism in this region of Colombia. Also, it is important to address whether this study uses 11 available microsatellites, which other studies have found using other genetic markers in this species? Other studies are mentioned in lines 336 to 340 and 359 to 371; what molecular data are used, and how can that be different from your results?

Lines 254 –255 and 290 - 291 indicate that population 2 is at a higher elevation than population 3, which seems incorrect since Population 3 has the two points with the highest elevation according to table 1 (Punte Quetame and Ibague (Tolima). Please clarify these values and the mean elevation for each detected population.

Point Medina Cundinamarca in Table 1 is cited as a Mexican point, but it is Colombian. Also, since Agua de Dios is divided into two populations, Table 1 should have two different coordinates and data depending on how the populations have been separated.

Lines 1 and 2. The title addresses environmental variation linked to population structure, which was not found in the results and is not present in any data analyses shown in the results.

Line 36: “...predominantly spread from bats to other species by the common vampire bat ...” This line is confusing and could target vampire bats or bats in general, but not both, as responsible for rabies at the same time.

Line 49: “Upon” feels awkward here – maybe “in”?

Line 60: “precipitous” does not fit here. Maybe “dramatic”?

Line 70 indicates that genetic diversity is linked to pathogen prevalence and virulence in wildlife, but does not indicate whether it is low or high genetic diversity. Also, it does have only a reference (Perez-Gonzales et al, 2021), I would recommend having more to support this statement.

Lines 81-83: This statement requires support. The two listed references are exclusively about climate and mention nothing about bat-borne emerging diseases. There are other papers that could support this, e.g., Olival et al. 2017 Nature, Brierly et al. 2015 Am Nat.

Line 220: When mentioning Fst, call it Wright’s Fst or just Fst throughout the whole text to avoid confusion.

Lines 222-225: This sentence does not make grammatical sense.

Lines 225-227: In contrast to what?

Lines 321 and 324 are convoluted and difficult to follow; also, values of genetic variation should be added to the paragraph.

Table 2: What is the right column of each metric? Why, for example, are the number of private alleles per locus in population 1 higher in the right column than the sum of the left? Is the left column an inter-site comparison and the right an inter-population comparison?

Table 2: Since the main comparison the authors wanted to make was elevation, can the elevations of these sites be added to Table 2 so the reader doesn’t have to flip back and forth, memorizing the names?

Figure 2: The PC graph needs axes. Which is PC1, which is PC2? What are the values of the eigenvalues? What percent of the variation do they explain?

Experimental design

The use of microsatellites seems appropriate given the need for as many informative loci as possible if other techniques, such as ddRAD or low-coverage genomes, were cost-prohibitive. As noted in the general reporting section, there are details lacking that make assessing their analyses difficult, especially the environmental variation.

Validity of the findings

Authors need to provide their raw data, report the environmental variation, and explain their spatial autocorrelation analyses.

Additional comments

Overall, we think this is useful information for the field and will yield fundamental data that will assist in understanding bat population structure and eventually disease spread. We strongly suggest the authors reframe their narrative to more accurately reflect the study they conducted, with a stronger focus on population structure and migration and less emphasis on zoonotic disease. And we require that they explain some of their analyses and data in greater detail. However, we are excited to see this become part of our knowledge of vampire bat population structure.

---

## Round 0.2 · accepted · Accept

· Academic Editor

Accept

Thanks for your thorough revisions and responses to the reviewer feedback.